# MOMENTUM BOOSTED EPISODIC MEMORY FOR IMPROVING LEARNING IN LONG-TAILED RL ENVIRONMENTS

## ABSTRACT

Conventional Reinforcement Learning (RL) algorithms assume the distribution of the data to be uniform or mostly uniform. However, this is not the case with most real-world applications like autonomous driving or in nature, where animals roam. Some objects are encountered frequently, and most of the remaining experiences occur rarely; the resulting distribution is called *Zipfian*. Taking inspiration from the theory of *complementary learning systems*, an architecture for learning from Zipfian distributions is proposed where long tail states are discovered in an unsupervised manner and states along with their recurrent activation are kept longer in episodic memory. The recurrent activations are then reinstated from episodic memory using a similarity search, giving weighted importance. The proposed architecture yields improved performance in a Zipfian task over conventional architectures. Our method outperforms IMPALA by a significant margin of 20.3% when maps/objects occur with a uniform distribution and by 50.2% on the rarest 20% of the distribution.

## 1 INTRODUCTION

Humans and animals roam around in environments that are unstructured in nature. However, existing algorithms in reinforcement learning are built around the assumption that environments are mostly uniform. Most of the time, a small subset of experiences frequently recur while many important experiences occur only rarely (Zipf, 2013; Smith et al., 2018). For example, imagine a deer trying to survive in an environment with predators. If it is drinking water from a source and it escapes from a potential predator, the deer cannot afford to learn slowly from multiple such experiences to learn to avoid dangerous places corresponding to water sources. It needs to learn that experience quickly and generalize across similar instances. Similarly, in autonomous driving, experiences are not uniform, and usually, the rare instances where there is an accident or unusual experiences are more critical in real-world settings. This is the fundamental premise on which the theory of complementary learning systems (McClelland et al., 1995; Kumaran et al., 2016) is proposed. In this framework, an intelligent agent needs to have a fast learning system and a slow learning system operating together to restructure the statistics of the environment for better survival internally and not be naive in expecting uniform environments. In the brain, this is hypothesized through the interplay between the hippocampus, a fast learning system, and the neocortex which is a slow learning system, and together they manage to generalize and retain experiences crucial to the goals of the organism (Kumaran et al., 2016). It is also true that to achieve their respective learning outcomes, both the systems need each other (Botvinick et al., 2019). The hippocampus is able to achieve fast learning through its reliance on the slow learning system of the cortex where high dimensional data coming from the sensory systems are converted to low dimensional representations which can be operated on by the hippocampus. Such top-down modulation from the cortex influences the processing in the hippocampus (Kumaran & Maguire, 2007). Similarly, the slow structured learning of the cortex happens through interleaved learning by replaying experiences stored in the hippocampus (O'Neill et al., 2010).

Here, we particularly look at this interplay between a fast learning and a slow learning system and apply this to solve the long-tailed phenomena. The reinforcement learning algorithm (Sutton & Barto, 2018b; Espeholt et al., 2018) uses the episodic buffer to generalize across experiences, and a

familiarity memory prioritizes long-tail data from the outputs generated by the RL algorithm. This prioritization of samples happens through a contrastive learning-related momentum loss which enables the unsupervised discovery of long-tailed data from the stream of experiences (Zhou et al., 2022). These prioritized samples are kept longer in memory and then, the hidden activations corresponding to these prioritized samples are reinstated in the recurrent layers of the RL network.

Our Main Contributions are:

- Proposing a first solution for the problem of navigating to objects occurring with a long tail distribution using deep reinforcement learning.
- Application of contrastive momentum loss for unsupervised discovery of long tail states in the context of reinforcement learning.
- Novel method to prioritize long-tail states in the buffer then reinstating hidden activations in recurrent layers.

## 2 BACKGROUND

### 2.1 MARKOV DECISION PROCESSES

Let's assume an environment $\mathcal{E}$ which provides the agent with an observation $S_t$, the agent selects an action $A_t$, and then the environment responds by providing the agent with the next state $S_{t+1}$. The interactions between the agent and environment are formalized by *MDPs* which are reinforcement learning tasks that satisfy Markov property (Sutton & Barto, 2018a). It is defined by the tuple $< \mathcal{S}, \mathcal{A}, \mathcal{R}, \mathcal{T}, \gamma >$ where, $\mathcal{S}$ represents the set of states, $\mathcal{A}$ is the set of actions, $\mathcal{R} : \mathcal{S} \times \mathcal{A} \to \mathbb{R}$ denotes the reward function. $\mathcal{T} : \mathcal{S} \times \mathcal{A} \to Dist(\mathcal{S})$ represents the transition function mapping state-action pairs to a distribution over next states $Dist(\mathcal{S})$ and $\gamma \in [0, 1]$ is the discount factor.

### 2.2 IMPALA

Previous works on the actor-critic framework have been based on a single learner multiple actor architectures where the communications are based on gradients with respect to policy parameters (Mnih et al., 2016b). IMPALA is a distributed off-policy actor-critic framework (Espeholt et al., 2018); here actors communicate a sequence of trajectories to the learners which give it a very high throughput. We consider IMPALA as our base architecture and build on top of it.

### 2.3 MEMORY SYSTEMS IN RL

Memory systems in humans allow them to retrieve the relevant set of experiences for decision-making in case of unseen circumstances. In neuroscience, some of the types of memories studied are - *Working Memory* and *Episodic Memory*. Working memory is short-term temporary storage while episodic memory is a non-parametric or semi-parametric long-term storage memory. Deep Reinforcement Learning agents with episodic memory, in particular a combination of non-parametric and parametric networks have shown improved sample efficiency and are suitable for decision-making in rare events. Blundell et al. (2016) used a non-parametric model to keep the best Q-values in tabular memory. Pritzel et al. (2017) in Neural Episodic Control proposed a differentiable-neural-dictionary to keep the representations and Q-values in a semi-tabular form. Hansen et al. (2018) took up a trajectory-centric approach to model such systems. Our work looks up to Fortunato et al. (2019)'s state-centric formulation of an Episodic Memory (MEM) which implements working memory using a latent recurrent neural network and an episodic memory.

### 2.4 SELF SUPERVISED LEARNING

Self Supervised Learning has been widely used in Reinforcement Learning to learn better representations for learning actions (Anand et al., 2019; Laskin et al., 2020). Here we look at the Contrastive Learning framework (Chopra et al., 2005; Chen et al., 2020; He et al., 2019) which uses similarity constraints to learn representations. The representation in pixel space is learned by bringing different views of the same images together and vice-versa. Usually, Self-supervised long-tailed learning methods have been developed mainly from a *loss perspective* or a *model perspective*. Focal loss

in hard-example mining (Lin et al., 2017; Liu et al., 2021) belongs to the former perspective of loss re-weighting. These methods are however limited by long tail sample discovery because they require labels to work. SSL from model perspective requires highly specific model design (Tian et al., 2021; Jiang et al., 2021). However, Zhou et al. (2022) takes up a different approach to long tail learning which is based on the data perspective. Zhang et al. (2021); Arpit et al. (2017); Feldman (2019) has studied the memorization effect in deep networks, and it has been observed that easy patterns are usually memorized before the hard patterns, and similar behavior can be observed in long-tail learning. Zhou et al. (2022) extends this by distinguishing between the long tail and head samples with respect to memorization in an unsupervised setting. They propose *Boosted Contrastive Learning* method which works with a momentum loss that enables the unsupervised discovery of long-tail data.

## 3 TASK

Our task is inspired by Zipf's Gridworld task in Chan et al. (2022). The task involves navigating to a target object in a partially observable gridworld environment. There are in total of 20 maps, and 20 objects in each of the maps are placed at random positions. The positions of these objects in these maps do not change during trials. The trial starts with the agent placed in the center. Given a target object, the agent has to find the object in the partially observable gridworld environment by taking at most 100 steps. If the object selected at the end is the correct object, the trial ends, and the agent gets a positive reward. If the object selected is incorrect or the number of steps exceeds the limit, the agent gets no reward.

Due to the unavailability of code for the environment during the time of our experimentation, we designed and implemented a similar Zipfian Gridworld environment with slight differences from the original environment proposed in Chan et al. (2022) but maintaining the core principles. Instead of embedding the target object in the top left corner of the view of the agent, we embed it in the top center with a black background. Also, we have just four actions (up, down, left, right) which can be used to navigate to any cell in the grid. It is ensured that all the objects in a map are distinct and the agent is able to reach any object present on the map using the set of four moves. An example of the agent's partial view can be seen in Figure 1a. There are a total of 10 maps, and in each of these maps, we have 10 objects. We chose the number 10 primarily because of the vast computational resources required to process 20 grid maps with 20 objects each.

$$zp(k, n, e) = \frac{1/k^e}{\sum_{i=1}^{n}(1/i^e)} \tag{1}$$

The probability of occurrence of the maps is governed by Zipf's power law (Equation 1), where $n$ is the number of maps/objects, $k$ is the map/object index $(1 <= k <= n)$ and $e$ is the Zipf's exponent. The same skew can be seen for target object selection in each map as shown in Figure 1b. To solve the task, the agent not only needs to explore the environment intelligently but also needs to memorize the path if the agent solves the trial correctly.

## 4 ARCHITECTURE

Given an image observation $(im)$, IMPALA's (Espeholt et al., 2018) feature extractor gives a pixel input embedding $(p)$ which is further passed to an LSTM network with the hidden state $(h)$ to get the new hidden state, policy, and value. As done in (Fortunato et al., 2019), we have a MEM module (orange-red buffer in Figure 3) that is used to find the memory $(m)$, which is then additionally fed as input to the LSTM network along with the pixel input embedding $p$. We only keep rare states in the MEM by introducing an additional 'familiarity' buffer (light cyan buffer in Figure 3) that uses boosted contrastive learning to prioritize and filter rare states for storing in MEM.

### 4.1 STATE FAMILIARITY USING BOOSTED CONTRASTIVE LEARNING

A 'familiarity' buffer is a circular buffer that contains states over which we can get a prediction on whether a state is rare or how rare the states are. To achieve this, we take inspiration from Zhou et al. (2022), where they improve performance on a long-tailed self-supervised learning task by proposing a momentum loss that can predict which samples among the dataset are long-tail samples.

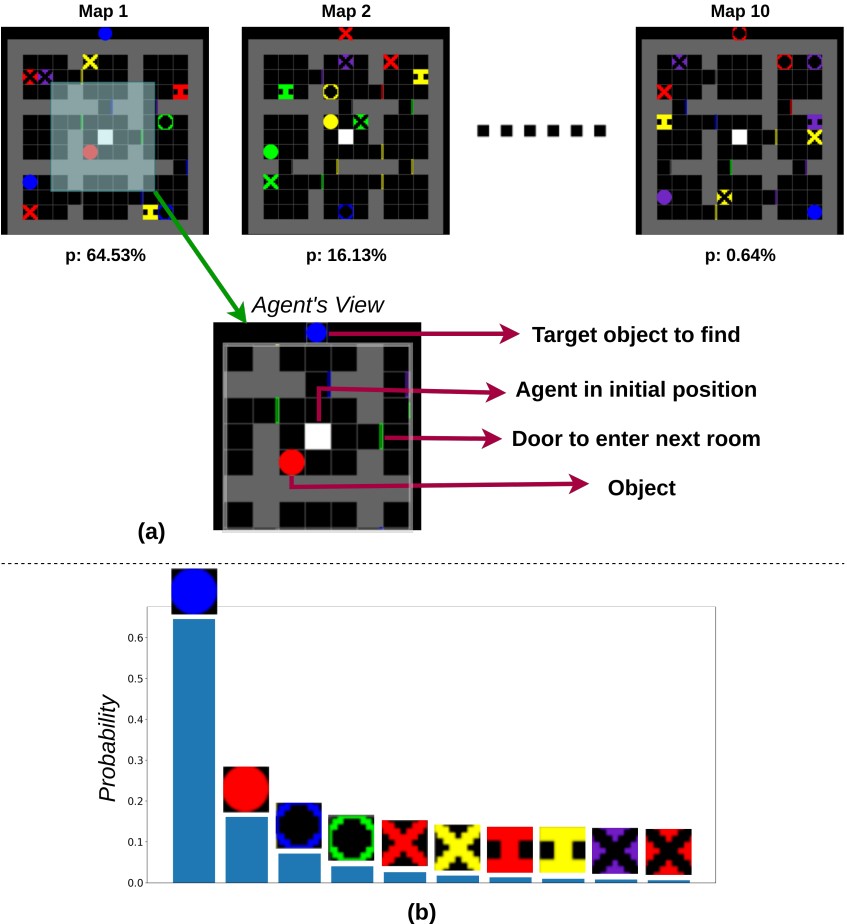

**(a)**

**(b)**

Figure 1: **Zipf's Gridworld Task:** Contains 10 maps, each with 10 objects placed at random locations. The location of these objects does not change during trials. The agent (white square) starts at the center in each trial and has to navigate towards the target object shown in the center at the top. The agent's view is shown in the bottom image in **(a)**. The target object to find, the agent's initial position, door, and object can be seen in the annotated image. The value 'p' below each map shows the probability of occurrence of the map in a trial, highlighting the skew in the distribution. A similar skew occurs for the distribution of objects in these maps. We can see this in the below part **(b)**, which shows the distribution of objects for the first map (most common). Adapted from Chan et al. (2022)

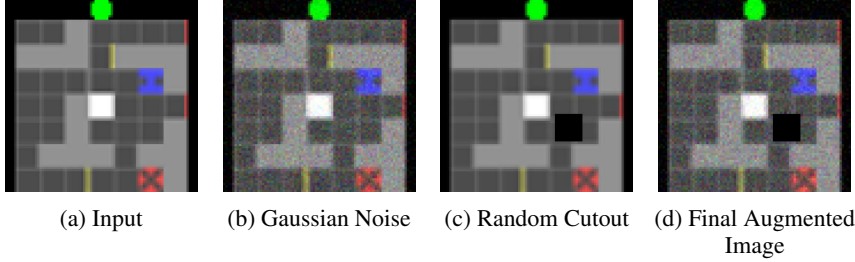

(a) Input  (b) Gaussian Noise  (c) Random Cutout  (d) Final Augmented Image

Figure 2: **Image augmentations for contrastive learning:** **(a)** Shows downsampled input image for a trial. **(b)** Input image after adding gaussian noise to it. **(c)** Input image after applying random cutout augmentation. The black rectangle near the agent's position is the area cutout. **(d)** Final augmented image after adding gaussian noise and random cutout.

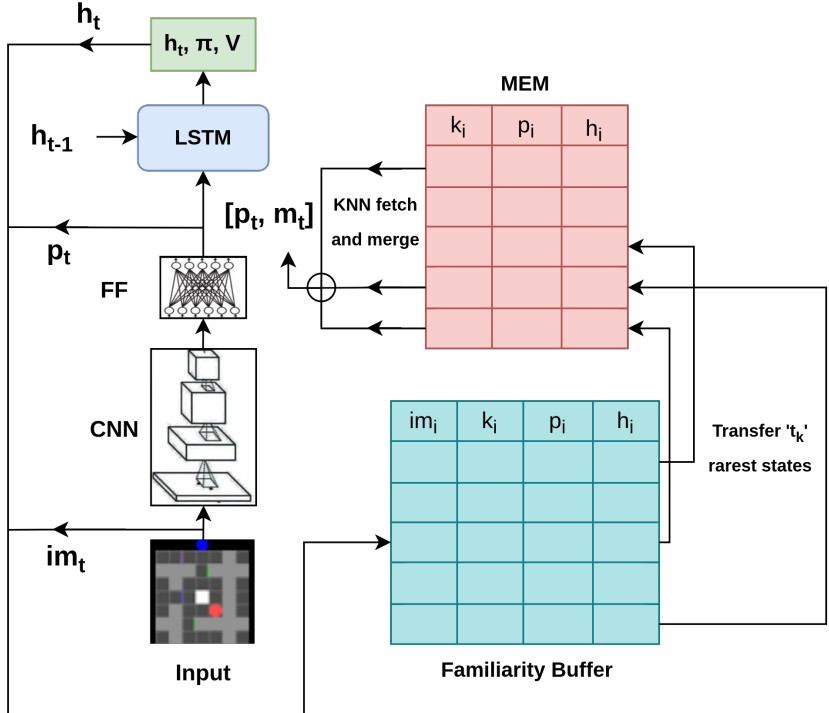

Figure 3: **Model Architecture:** The figure shows our momentum-boosted episodic memory architecture pipeline. The IMPALA backbone consists of a CNN feature extractor followed by a Feed Forward layer that gives the embedding. This embedding is concatenated with the one hot action encoding, reward & memory to get pixel embedding $p_i$ and then given to the LSTM network for further processing with working memory. The LSTM network additionally takes the past hidden state $h_{t-1}$ as input. During training, the input image, pixel embedding, LSTM hidden states, and keys are stored in the familiarity buffer. The momentum loss tracked on this buffer during contrastive learning is then used to prioritize long-tail states. The MEM is then periodically updated with top $t_f$ states from the familiarity buffer. The memory ($m_t$) is computed from the MEM using a weighted sum ($\bigoplus$) after fetching using a KNN similarity search on the keys present in the MEM (Equation 8).

Our 'familiarity' buffer (light cyan buffer in Figure 3) contains the input image ($im$), key ($k$), pixel input embedding ($p$) and LSTM hidden state ($h$). We will define these terms in the following section. In each learning step of IMPALA (Espeholt et al., 2018), the batch of trajectories is sent to this buffer. The entire trajectory isn't sent to the buffer, but a subset of it (states at a hop of $hp$ from the start of the trajectory) is sent. For a trajectory $\text{Tr} = (s_1, s_2, ..., s_k)$ of size $k$ consisting of states $s_i$, the subset of trajectory is defined as

$$\text{Tr}_{\text{subset}} = \left( s_1, s_{(1+hp)}, ..., s_{\left(1 + \lfloor \frac{k-1}{hp} \rfloor * hp \right)} \right) \tag{2}$$

The same feature extractor of IMPALA is trained using an additional auxiliary contrastive loss (Equation 6) on states present in the familiarity buffer. Once the circular buffer is full of states, the contrastive learning process starts to help the learning process. For each image sample $im_i$ in the buffer, we find its pixel input embedding $p_i$ after IMPALA's feature extractor. The same feature extractor is used on the augmented image to get another embedding $p_i^{aug}$. The image $im_i$ is augmented using two augmentations, namely gaussian noise (Boyat & Joshi, 2015) and random cutout (DeVries & Taylor, 2017). We use these two augmentations because they are simple and don't affect the task. For example, using something like random conv (Laskin et al., 2020) will change the color of all objects in the observation, and this might result in the agent confusing it to be an image (observation) from some other trial. For adding gaussian noise to the image, we first generate an image of the same dimension as that of the original image that is filled with random numbers from a

normal distribution with mean 0 and variance 1. This image is then added to the original image after amplifying by a factor of $sigma$. For random cutout, we take a random location in the image and cut a rectangular area of random size. By cutting, we mean replacing the pixels in that rectangular region with black pixels.

During the training process, we consider the embeddings $p_i$ and $p_i^{aug}$ as positive pairs for contrastive learning. The NT-Xent loss ( Sohn (2016), Equation  6) is used to calculate the loss per sample. For each sample $i$ in the circular buffer, we track its momentum loss following  Zhou et al. (2022). The momentum loss assists in knowing which samples in the circular buffer are long-tail samples. For a sample $i$, if for $T$ consecutive epochs the contrastive losses are $\{\ell_{1,i}^T, \ell_{2,i}^T, ..., \ell_{T,i}^T\}$, then the moving average momentum loss is defined as follows:

$$\ell m_{i,1}^m = \ell_{1,i}^T; \;\; \ell m_{i,t}^m = \beta \ell m_{i,t-1}^m + (1-\beta)\ell_{t,i}^T \tag{3}$$

where $\beta$ is a hyperparameter that controls the degree smoothed by the historical losses. The final normalized momentum used to find the familiarity of states is defined as

$$M_{i,t} = \frac{1}{2}\left(\frac{\ell m_{i,t}^m - \bar{\ell m}_t^m}{max\{|\ell m_{j,t}^m - \bar{\ell m}_t^m|\}_{j=1,...,N}} + 1\right) \tag{4}$$

where $\bar{\ell m}_t^m$ is the average momentum loss of the dataset at the $t^{th}$ training step of the algorithm and N is the number of samples. The higher the momentum value $M_{i,t}$, the higher is the rareness of the sample in the circular buffer. The model is trained end to end by optimizing both IMPALA's loss and the auxiliary contrastive loss. Let the loss given by IMPALA be $\mathcal{L}_{\text{impala}}$ and that given by the contrastive learning branch be $\mathcal{L}_{\text{contrastive}}$, then we define the final loss to be the one shown in Equation  5 below.

$$\mathcal{L} = \mathcal{L}_{\text{impala}} + \gamma * \mathcal{L}_{\text{contrastive}} \tag{5}$$

where $\gamma$ is a hyperparameter. The contrastive loss is given by:

$$\mathcal{L}_{\text{contrastive}} = \frac{1}{N}\sum_{i=1}^{N} -\log\frac{\exp\left(\frac{p_i^\top \cdot p_i^{aug}}{\tau}\right)}{\sum_{p_i' \in X'}\exp\left(\frac{p_i^\top \cdot p_i'}{\tau}\right)} \tag{6}$$

where N is the number of samples, $X'$ represents $X^- \cup \{p_i^{aug}\}$, $(p_i, p_i^{aug})$ is the positive sample sample pair, $X^-$ is the negative sample set of $p$ and $\tau$ is the temperature.

## 4.2 Combining Familiarity with Episodic Memory

The base architecture we use is IMPALA with MEM proposed in  Fortunato et al. (2019). Note that we do not use the one-step contrastive predictive coding (CPC) module they introduced. In the paper, an Episodic Memory (MEM) is introduced on top of the IMPALA architecture, which is a circular buffer that stores the pixel embedding ($p$), LSTM hidden state ($h$), and the key ($k$), which is calculated using

$$k = W[p, h] + b \tag{7}$$

where W and b are learnable parameters and $[p, h]$ denotes the concatenation of $p$ and $h$ along the dimension axis. When IMPALA+MEM comes across situations with a similar context, it uses a method to learn how to save summaries of previous experiences and extract crucial data. The neural network (controller), which generates the model predictions, also receives extra inputs from the readings from memory. Learning long-term dependencies, which can be challenging when depending solely on backpropagation in recurrent architectures, is made simpler by successfully enhancing the controller's working memory capacity with experiences from various time scales received from the MEM. In the original IMPALA+MEM algorithm, the pixel embeddings $p_i$, LSTM hidden states $h_i$, and keys $k_i$ were computed after the feature extraction layer of IMPALA and then added to the MEM buffer. But, we instead add this to the familiarity buffer, which computes the familiarity of states based on the momentum loss and then periodically sends it to the MEM buffer to facilitate learning during rare trials. The agent is able to perform well on frequent states even if the MEM module is not used and only the simple IMPALA algorithm is used. Hence, we only maintain relatively rare states in the MEM buffer to help learn on rare states that actually require the help of an external episodic memory module. The MEM gets $t_k$ most rare states from the familiarity buffer after every $t_f$ training epochs of contrastive learning, where $t_k$ and $t_f$ are hyperparameters.

The overall architecture can be seen in Figure 3. For a stimulus $p_t$ and previous hidden state $h_{t-1}$, the agent chooses the most pertinent events to give as input $m_t$ to the LSTM network using a type of dot-product attention (Bahdanau et al., 2015) over its MEM. Using the key $k_t$, formed by $p_t$ and $h_{t-1}$ using Equation 7, a K Nearest Neighbour (KNN) search is done over the keys in MEM to find the most relevant $K$ keys. The hidden states for these $K$ relevant items are combined using the below-weighted sum (Equation 8) to get additional input $m_t$ to be given to the LSTM network.

$$m_t = \frac{\sum_{i=1}^{K} w_i h_i}{\sum_{i=1}^{K} w_i} \qquad w_i = \frac{1}{||k_t - W[p_i, h_i] - b||_2^2 + \epsilon} \tag{8}$$

where $\epsilon$ is a small constant and $||x||_2^2$ represents the squared $L2$ norm of $x$.

---

**Algorithm 1** Pseudocode for our algorithm

---

1: **Inputs:**
    $fm, mem$         // Familiarity memory and MEM
    $t_f$                 // Transfer frequency
    $t_k$                 // Number of rare instances to transfer
    $T$                 // Number of IMPALA training epochs
    $hp$              // Within trajectory hop
    $\gamma$                 // Contrastive loss weight
2: **Initialize:**
    $fm.buffer \leftarrow \{\}$    // Clear FM buffer
    $mem.buffer \leftarrow \{\}$ // Clear MEM buffer
3: **for** t = 1 to T **do**
4:     $trajectories \leftarrow get\_impala\_batch(t)$
5:     $im, k, p, h, impala\_loss \leftarrow impala\_train(trajectories, hp, mem)$
6:     $fm.add(im, k, p, h)$                        // subset added according to Equation 2
7:     $contrastive\_loss \leftarrow fm.contrastive\_train()$      // loss calc. as shown in Equation 6
8:     **if** $t\%t_f == 0$ **then**
9:         $momentum\_values \leftarrow fm.calculate\_normalized\_momentum()$    // Equation 4
10:        $rare\_experiences \leftarrow fm.get\_rare\_k(t_k, momentum\_values)$
11:       $mem.add(rare\_experiences)$
12:     $final\_loss \leftarrow impala\_loss + \gamma * contrastive\_loss$            // Equation 5
13:     $final\_loss.backward()$

---

## 5   EXPERIMENTS AND RESULTS

We compare the results of our method with mainly four different types of architecture. The first architecture that we compare is IMPALA (Espeholt et al., 2018), which is an off-policy actor-critic framework and has shown substantial improvements over baselines like Clemente et al. (2017); Mnih et al. (2016a). The second one is the IMPALA+MEM introduced in Fortunato et al. (2019). The third experiment that we do is IMPALA with visual reconstruction. Chan et al. (2022) have experimented with visual reconstruction (Hill et al., 2021), and it has been seen that it helps in improving performance on rare trials by a small margin (see the second row of Table 1). In this, we add an extra task for visual reconstruction on top of IMPALA with a CNN-based autoencoder. The fourth one is where we included contrastive learning to learn good embeddings. It is to be noted that this approach (see the fourth row of Table 1) does not find the rare states in the familiarity buffer, it samples $k$ states uniformly randomly from the familiarity buffer. In contrast in the proposed approach, we pass the rare $k$ states to MEM from the familiarity buffer (see the last row of results in Table 1). From these experiments, we can see clearly that contrastive learning (feature representation learning) alone cannot give good performance and we also need to add a familiarity buffer to prioritize rare states and pass them on to MEM. The training curves (Training/Zipfian Accuracy) and different ablations can be seen in Section A.2. We can see that the Zipfian mean episode return for our method increases faster than all the other methods in the initial phase of training and also converges later to have the highest accuracy ($\approx$99.9%). The hyperparameters used for our model are listed in Section A.3. The results on Deepmind Zipf's Gridworld are shown in Section A.1.

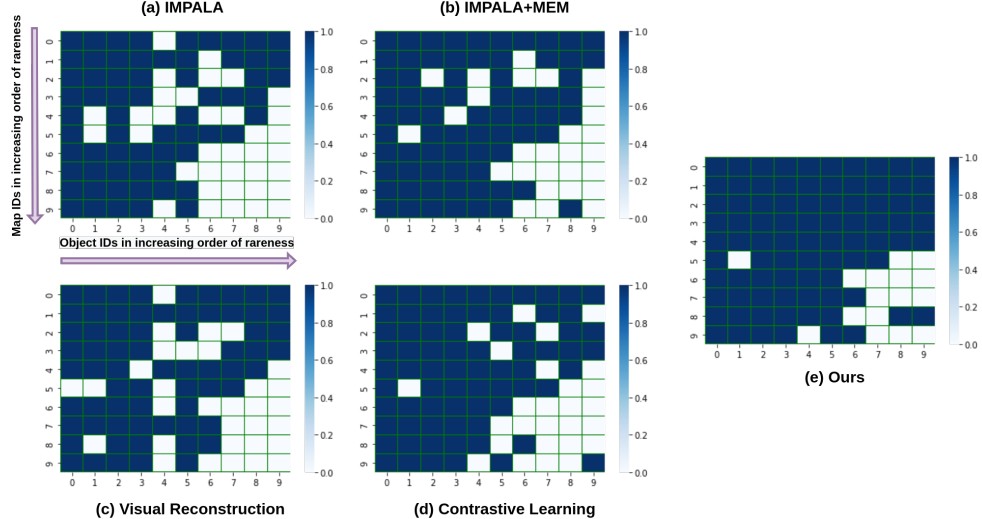

Figure 4: **Performance plots:** **(a)** Performance of IMPALA agent on each map and object. The y-axis denotes the map axis, and the x-axis denotes the object axis. Value at (i, j) shows the performance (0-1 scale) of the agent on the trial where the object with ID j is chosen at the map with ID i. An increase in i and j means an increase in the rareness of the map and object respectively according to the Zipf's distribution (Equation 1). **(b)** Performance of IMPALA with MEM added. We can see that are some medium-rare trials in which the agent has learned to navigate and learn the task. **(c)** IMPALA with Visual Reconstruction using CNN-based autoencoder. **(d)** Performance of IMPALA+MEM with contrastive learning. **(e)** Performance of our agent consisting of familiarity buffer that highlights long tail samples for MEM using modified boosted contrastive learning.

Table 1: **Evaluation Performance:** We compare four different methods with our algorithm namely IMPALA, IMPALA+Visual Reconstruction using a simple CNN-based autoencoder, IMPALA+MEM, and IMPALA+MEM with only contrastive learning. We report median results across three runs ($\pm$ absolute median deviation across runs) with distinct random seeds for models trained for $4 \times 10^7$ steps. Our method (IMPALA+MEM+Contrastive Learning+Rare State Prioritization using Familiarity Buffer) beats the remaining methods by a large margin on all three evaluation metrics.

|  | Accuracy (%) | | |
|---|---|---|---|
| Method | Zipfian | Uniform (All maps and objects) | Rare (Rarest 20% objects on rarest 20% maps) |
| IMPALA | $95.0 \pm 0.03$ | $64.1 \pm 0.02$ | $0 \pm 0.00$ |
| IMPALA + Vis recon | $95.9 \pm 0.02$ | $68.0 \pm 0.01$ | $0 \pm 0.00$ |
| IMPALA + MEM | $97.2 \pm 0.03$ | $72.2 \pm 0.04$ | $25.1 \pm 0.01$ |
| IMPALA + MEM + Contr. Learning | $98.1 \pm 0.02$ | $73.1 \pm 0.01$ | $25.3 \pm 0.02$ |
| **Ours** | $\mathbf{99.9 \pm 0.01}$ | $\mathbf{84.4 \pm 0.01}$ | $\mathbf{50.2 \pm 0.02}$ |

Having a higher training accuracy is not what we are looking for; instead, we want to have good accuracy when tested uniformly or in rare instances. Figures 4(a-e) show the performance of the five architectures on all the maps and objects of our environment. For each (map, object) combination, we plot the average performance across 50 trials. Figure 4a shows the performance of the IMPALA agent. We can see that the agent is not able to learn the extreme rare trials and also some medium-rare trials are not learned by the agent. Figures 4 (b) & (d) show the performance of IMPALA+MEM with and without contrastive learning respectively. In the case of IMPALA+MEM with contrastive

learning, the familiarity buffer is sampled uniformly randomly to fetch states for the MEM. We see that the performance is almost similar, but still, the medium-rare trials are not learned. Figure 4c shows the performance of IMPALA with added visual reconstruction using CNN based autoencoder. This performs slightly better than the baseline (IMPALA) but fails to match the performance of other agents. Lastly, in Figure 4e we can see the performance of our agent with the familiarity module. The medium-rare trials are being learned, and also some very rare trials are been done successfully by our agent. Table 1 gives more insights into our agent's performance. Our agent is able to beat other compared agents by a large margin on all of the three evaluation metrics (Training/Zipfian accuracy, Uniform accuracy, and Rare accuracy).

## 6 DISCUSSION

This paper deals with the problem of long-tailed distribution in reinforcement learning. Inspired by the theory of complementary systems, which states that an intelligent agent requires a fast and slow learning system acting complementary to each other. Here, the momentum loss of contrastive learning provides a mechanism to detect long tail samples in an unsupervised manner. These samples are then prioritized to be stored in a separate buffer that stores hidden activation associated with such states. When a rare sample comes, a similarity search is done to find relevant keys, and the corresponding hidden activations are merged to be reinstated in the recurrent layers.

This architecture relates to how the hippocampus which is a fast learning system acts in tandem with the slow learning cortex of the organism to store relevant experiences and replay them to overcome the statistics of the environment the organism is subjected to (O'Neill et al., 2010). The episodic memory relies on the network to discover long tail data from the incoming data stream. The network relies on the episodic memory for identifying the relevant memory of the long tail data in order to reinstate it in the recurrent layers of the working memory system to execute the episodic sequence. Similarly, the brain could reinstate episodic sequences from the hippocampus to the working memory when animals execute a task.

Furthermore, dopamine neurotransmitter has been found to detect novel states and relay them to the hippocampus (Duszkiewicz et al., 2019). The firing of dopamine neurons is also related to curiosity and learning progress which is analogous to momentum loss here (Ten et al., 2021; Gruber & Ranganath, 2019). The neural network prioritizes its representations and persistence of memory based on how informative the state is for executing the task. This architecture can also potentially execute episodic generalization as proposed by Fortunato et al. (2019) but adapted for long-tailed data. The resulting embedding could be different, but the underlying dynamics of executing the task could be the same. Future work could look at how we could extend this to more realistic 3D environments and include systematic generalization to compose temporally separated rare states to solve tasks (Fodor & Pylyshyn, 1988).

## 7 CONCLUSION

This paper attempts to overcome the problem of long-tailed data for reinforcement learning which conventional architectures do not address well owing to their underlying assumptions. An unsupervised long-tail discovery method using self-supervised momentum loss is used to prioritize long-tail data. Using this prioritization, an episodic storing of hidden activations is done to be later reinstated in the recurrent layers so that rare trajectories are executed. Both of these proposed features are crucial in enabling the network to perform better than conventional architectures on a long-tail dataset. We hope this work will encourage the development of new RL methods in such data distributions and finally enable the development of agents capable of learning from a lifetime of non-uniform experience.

## 8 ACKNOWLEDGEMENT

To be added later.

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

## A  APPENDIX

### A.1  RESULTS ON DEEPMIND'S GRIDWORLD

We evaluate various baselines on Deepmind Zip's Gridworld Environment (10 maps and 10 objects) and report results in Table 2.

Table 2: **Evaluation Performance:** Deepmind Zipf's Gridworld Environment.

| | Accuracy (%) | | |
|---|---|---|---|
| Method | Zipfian | Uniform (All maps and objects) | Rare (Rarest 20% objects on rarest 20% maps) |
| IMPALA | $88.3 \pm 0.02$ | $41.1 \pm 0.01$ | $0 \pm 0.00$ |
| IMPALA + Vis recon | $90.2 \pm 0.02$ | $45.9 \pm 0.01$ | $0 \pm 0.00$ |
| IMPALA + MEM | $92.9 \pm 0.03$ | $51.2 \pm 0.02$ | $25.0 \pm 0.02$ |
| IMPALA + MEM + Contr. Learning | $94.8 \pm 0.02$ | $52.3 \pm 0.01$ | $25.1 \pm 0.02$ |
| Ours | $98.5 \pm 0.02$ | $66.3 \pm 0.01$ | $25.2 \pm 0.01$ |

### A.2  SUPPLEMENTARY ANALYSES

The training curve for the main experiments can be seen in Figure 5.

The effects of changing different hyperparameters of our architecture are explained here.

We look at the 'K' value of K Nearest Neighbour, Trajectory Hop ($hp$), Rare State Transfer amount ($t_k$), and Rare State Transfer Frequency ($t_f$) for our ablation study. The effect on the training of these hyperparameters can be seen in Figure 6.

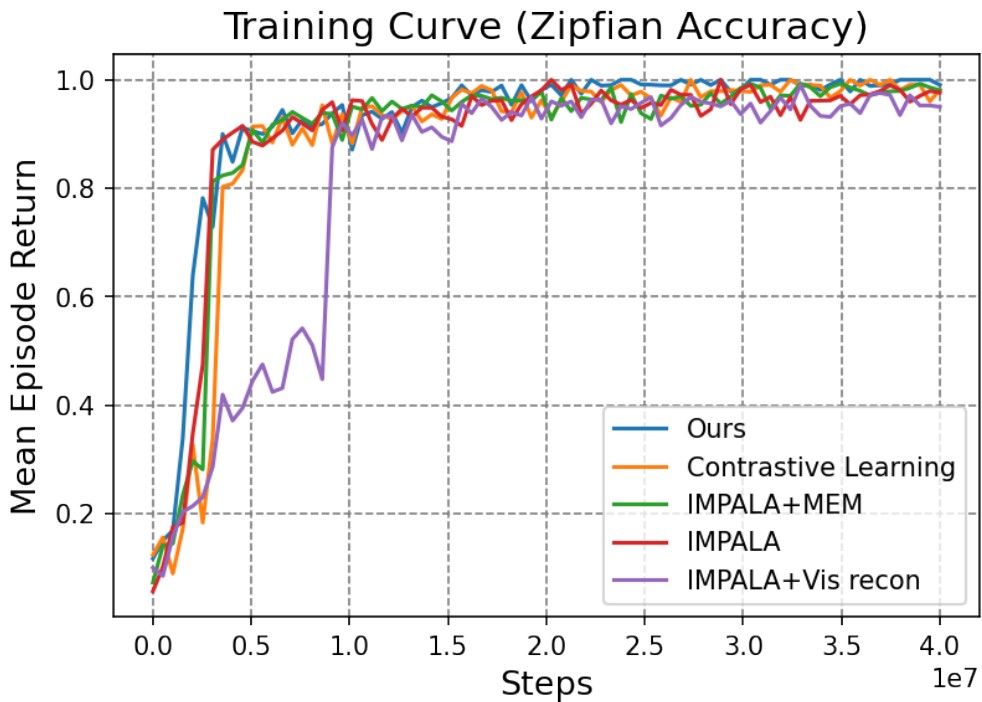

Figure 5: **Performance plots:** Training curves for different experiments.

Table 3: Effect of KNN 'K' value.

| K Value | Accuracy (%) | | |
| | Zipfian | Uniform (All maps and objects) | Rare (Rarest 20% objects on rarest 20% maps) |
| --- | --- | --- | --- |
| 2 | 99.4 | 79.0 | 0 |
| 8 | 98.5 | 67.6 | 0 |
| 16 | 99.9 | 84.4 | 49.9 |
| 32 | 97.3 | 74.9 | 24.9 |

Table 4: Effect of trajectory hop $(hp)$.

| $(hp)$ | Accuracy (%) | | |
| | Zipfian | Uniform (All maps and objects) | Rare (Rarest 20% objects on rarest 20% maps) |
| --- | --- | --- | --- |
| 2 | 99.1 | 78.4 | 24.8 |
| 8 | 99.3 | 79.1 | 24.9 |
| 16 | 99.9 | 84.4 | 49.9 |
| 32 | 99.1 | 77.7 | 25.2 |

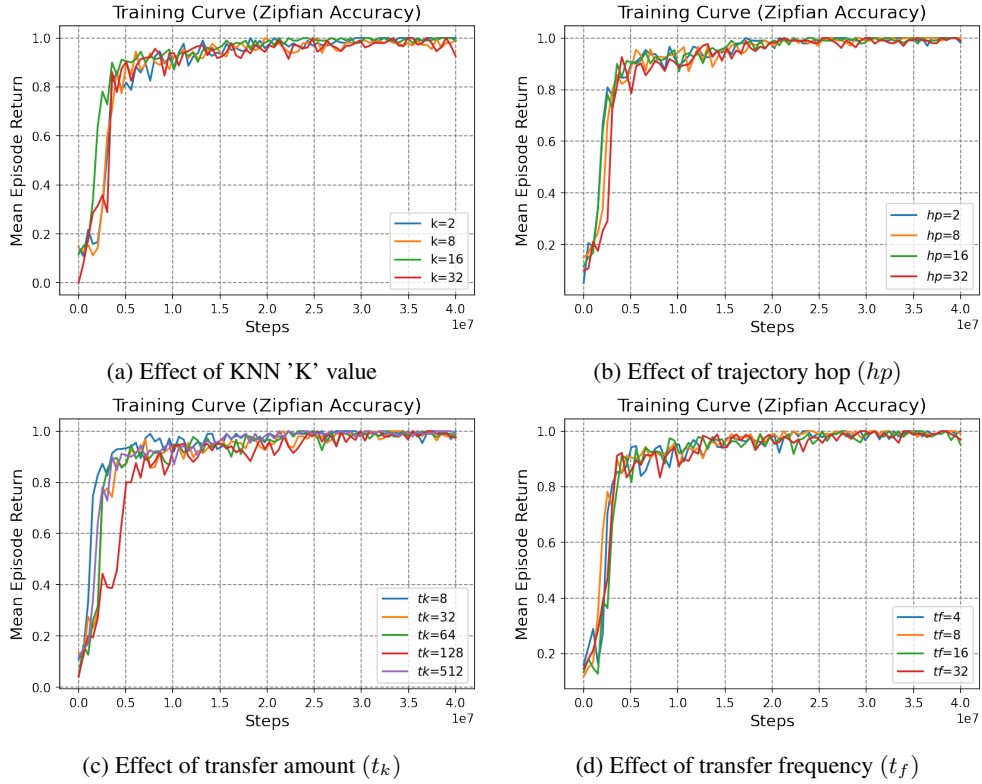

Figure 6: **Training Performance:** Effect of changing different hyperparameters on training.

Table 5: Effect of transfer amount $(t_k)$.

| $(t_k)$ | Zipfian | Uniform (All maps and objects) | Rare (Rarest 20% objects on rarest 20% maps) |
|---|---|---|---|
| | | Accuracy (%) | |
| 8 | 99.2 | 81.1 | 24.9 |
| 16 | 99.9 | 84.4 | 49.8 |
| 32 | 99.2 | 76.0 | 0 |
| 64 | 98.9 | 77.9 | 0 |
| 128 | 98.7 | 69.8 | 0 |

Table 6: Effect of transfer frequency $(t_f)$.

| $(t_f)$ | Zipfian | Uniform (All maps and objects) | Rare (Rarest 20% objects on rarest 20% maps) |
|---|---|---|---|
| | | Accuracy (%) | |
| 4 | 99.1 | 79.9 | 25.0 |
| 8 | 99.9 | 84.4 | 49.8 |
| 16 | 99.3 | 82.0 | 49.9 |
| 32 | 99.1 | 76.9 | 74.8 |

## A.3 EXPERIMENT HYPERPARAMETERS

|  | Zipf's Gridworld |
|---|---|
| Image Width | 84 |
| Image Height | 84 |
| Action Repeats | 1 |
| Unroll Length | 32 |
| Discount ($\gamma$) | 0.99 |
| Baseline loss scaling | 0.5 |
| Entropy cost | 0.01 |
| Action Repeats | 1 |
| Optimizer | RMSProp |
| Learning rate | $3e-4$ |
| Number of training steps | $4e7$ |
| Maximum steps in a trial | 100 |

|  | Additional Parameters |
|---|---|
| Zipf's Exponent ($e$) | 2 |
| Number of Actors | 50 |
| Trajectory Hop ($hp$) | 16 |
| Average Momentum Beta ($\beta$) | 0.97 |
| Loss Gamma ($\gamma$) | 0.5 |
| MEM Buffer capacity | 1024 |
| Familiarity Memory Buffer capacity | 1024 |
| Rare State Transfer Amount ($t_k$) | 512 |
| Rare State Transfer Frequency ($t_f$) | 8 |
| KNN ($K$) | 16 |
| Epsilon ($\epsilon$) | $1e-3$ |
| Sigma ($sigma$) | 0.05 |

