# OpenReview forum: "Momentum Boosted Episodic Memory for Improving Learning in Long-Tailed RL Environments"
_ICLR.cc/2023/Conference — Submitted to ICLR 2023_

### Official Review · Reviewer_AqsP · 2022-10-20

**Confidence:** 4
**Correctness:** 3
**Technical Novelty And Significance:** 3
**Empirical Novelty And Significance:** 2
**Recommendation:** 5

**Clarity, Quality, Novelty And Reproducibility:**

The paper is moderately clear; I think the presentation could be improved in some places by adding more signposting to guide the reader. For example, at the beginning of section 4, adding a short paragraph to describe the pieces of the method and how they fit together would help frame the rest of the section.

The quality seems decent, though there are weaknesses as noted above.

The paper seems fairly novel—I have not seen similar approaches applied to long-tailed problems in RL.

**Strength And Weaknesses:**

Strengths:
The idea is interesting and compelling.
The results on the environment considered are impressive.

Weaknesses:
Using only a single environment makes the results somewhat limited. It would be nice to see experiments on another domain, to verify whether the results generalize.
It would be very useful to consider more ablations to the procedure.
Is the momentum in the familiarity necessary, or would using a simple average, or even just a single value of the loss give similar performance? While Zhou et al. presumably explored this in the image setting, it’s not clear that the results would be the same in RL.
Why these particular data augmentations, as opposed to others e.g. the others used in methods like SimCLR [Chen et al., 2020]?

Comments / notes:
The idea of identifying state familiarity through loss on an unsupervised method on those states is related to ideas that have been used to measure state novelty in exploration, such as in RND [Burda et al., 2018]. It might be worth making this link in the paper, because it suggests the possibility of more fertile overlap between these seemingly-disparate research streams in the future.
“Due to the unavailability of the code for the environment …” — The code for Zipf’s Gridworld task has been released (at least now) at https://github.com/deepmind/zipfian_environments/tree/main/gridworld — in fact, I’m using it in some present work. Might be worth updating this comment. The other environments could potentially be used to address the weakness noted above.

References
------

Burda, Y., Edwards, H., Storkey, A., & Klimov, O. (2018, September). Exploration by random network distillation. In International Conference on Learning Representations.

Ting Chen, Simon Kornblith, Mohammad Norouzi, Geoffrey Hinton Proceedings of the 37th International Conference on Machine Learning, PMLR 119:1597-1607, 2020.


**Summary Of The Paper:**

This paper proposes a method for tackling RL with long-tailed distributions. It uses a self-supervised contrastive method to estimate state familiarity, and then adds the rare states to an episodic memory module based on prior work. The paper considers one task domain, in which it demonstrates compelling improvements from the proposed approach.

**Summary Of The Review:**


I think this paper offers an interesting idea and some compelling preliminary results. I think it would be a great workshop paper now, but I feel it would need experiments on one more domain and some more ablations to accept as a main conference paper.

---

> ### Author Response · Authors · 2022-11-16
> **Response to Reviewer AqsP**
>
> The authors thank the reviewers for insightful feedback that helped improve our work. Please find our response to each comment below.
>
> **Q1: Using only a single environment makes the results somewhat limited. It would be nice to see experiments on another domain, to verify whether the results generalize.**
>
> There were 2 more tasks introduced in [1], but we didn't consider them for the following reasons:
>
> - The Zipf’s Labyrinth task tests the ability of the agents to learn in a meta-learning setup. Our problem is focused on learning in a single environment where the experiences are skewed.
> - The Zipf’s playroom tests the ability of the agent to learn in an environment where the agent is supplied with a string instruction, and it has to lift/put the corresponding object. This task mainly focuses on the ability to map language input to the corresponding objects in a skewed setting. Our architecture doesn’t take language as input but takes the instruction (target object to choose) along with the pixel observation (i.e., the target object is embedded in the RGB input).
>
> We could try this on some Atari Games, but we haven’t seen any such task where this long-tailed skewness is visible and well-highlighted. The main reason we used Zipf's gridworld is that it tests the agent on extreme conditions where the experiences can be very skewed. For example, the rarest 20% trials occur with a probability of less than 0.00006346. The skewness in Atari Games is not that pronounced, hence it won’t be fair the judge the agents based on the metrics that are used there (there won’t be much difference in the accuracy).
>
> We can design some new long-tail tasks and check the performance of our method on them, maybe in a 3D setting like Psychlab. We shall plan this for the next iteration of our paper.
>
> **Q2: Is the momentum in the familiarity necessary, or would using a simple average, or even just a single value of the loss give similar performance? While Zhou et al. presumably explored this in the image setting, it’s not clear that the results would be the same in RL.**
>
> We tried single value of loss, but the results were worse than the one with momentum. This is because the momentum additionally captures the change in the contrastive losses, which tells how fast/well it is able to learn on those samples.
>
> **Q3: Why these particular data augmentations, as opposed to others e.g. the others used in methods like SimCLR [Chen et al., 2020]?**
>
> We use these two augmentations because they are simple and don't affect the task. For example, using something like Color distort [1] will change the color of all objects in the observation, and this might result in the agent confusing it to be an image (observation) from some other trial.
>
> **Q4:  “Due to the unavailability of the code for the environment …” — The code for Zipf’s Gridworld task has been released (at least now) at https://github.com/deepmind/zipfian_environments/tree/main/gridworld — in fact, I’m using it in some present work. Might be worth updating this comment. The other environments could potentially be used to address the weakness noted above.**
>
> We didn't use the other two tasks in [2] due to the reasons mentioned in **Q1**. The code for the environments introduced in [2] wasn't available online when we started doing our experiments. It got released when we were midway through our ablation experiments. We didn't want to wait for the code release and hence had to resort to implementing our own version of this task.
>
> Now that the original gridworld environment is available, it's not fair if we don't do experiments on that. Hence, we have now reported results for that as well (Supplementary Section A.1) in the updated version of the paper. We would like to thank the author for pointing this out.
>
> [1] Chen et al., "A Simple Framework for Contrastive Learning of Visual Representations"
>
> [2] Chan et al., "Zipfian environments for Reinforcement Learning."

---

> > ### Comment · Reviewer_AqsP · 2022-11-16
> > **Thanks for the clarifications!**
> >
> > Thanks to the authors for the clarifications; I do find the work quite interesting and I look forward to reading future versions of the paper.

---

### Official Review · Reviewer_cQdQ · 2022-10-23

**Confidence:** 4
**Correctness:** 2
**Technical Novelty And Significance:** 2
**Empirical Novelty And Significance:** 3
**Recommendation:** 5

**Clarity, Quality, Novelty And Reproducibility:**

The paper is clear for the most part, although Section 4 needs some further work:
* It should clarify how sensible is the mthod to the different hyperparameters
*End of page 3, it should give a reference to the contrastive loss since it hasn't been introduced yet
* Page 4 line 3, "the augmented image" hasn't been explained yet
* It would be good to give some intuition there on why are you augmenting the image by adding noise and cutting a region
* In section 4.1 you explain the momentum Eq. 4 and the contrastive loss (Eq.5 and 6). But not how they are related, which makes it confusing
*Eq.7, you never explain what is v there
*Some intuition on equation 8 is missing too.
* Additionally, why using KNN and not the classic query-key match with hard attention? Hard attention is classically used in that context so would be good to motivate why authors did differently

Quality is a fair for the basic points, but I have major concerns due to the points I raised above about the method explanation and the lack of ablations and at least one baseline that is not the vanilla method authors are working with.

The novelty of the contributions feel incremental but fair.

Authors give plenty of details for reproducibility in the appendix although there is no mention for the code



**Strength And Weaknesses:**

Authors make a great work introducing the topic of research motivations and goals. Empirical results evidence that the presented method boosts the ability of agents to track long-tailed events.

I believe there are two major weaknesses in this work.

First, is the "method" section (Sec. 4), which is sometimes disjointed and could also improve with giving some intuition at some points. It also presents plenty of hyperparameters for the method, without detailing much about how sensible the proposed solution is to the hyperparameters. I'll detail further in the box below

Second, the experiment section is too narrow. In the sense that:
* It only contrast their extension of IMPALA+MEM against IMPALA and IMPALA+MEM, it would have been a great addition to compare with a completely different kind of memory-based solution like HCAM [1] to contrast. It would also greatly impact the significance of this work if authors demonstrated that the familiarity buffer can be integrated with other methods than MEM
* It would really benefit from ablation experiments. This would let the community know how influential is each element of your proposed solution, also it would clarify how sensible it is to the different hyperparameters.

[1] Lampinen, Andrew, et al. "Towards mental time travel: a hierarchical memory for reinforcement learning agents." Advances in Neural Information Processing Systems 34 (2021): 28182-28195.

**Summary Of The Paper:**

This works presents a memory based method to deal with learning in Zipfian distributions, i.e., when some of the training data is rarely visited. To do so, authors start from an existing IMPALA + an Episodic Memory (MEM) method, and introduce a "familiarity buffer" that acts as a long-term memory and selects the top k rarest events to send them to the MEM, that acts like a shorter-spam or working memory.

Experiments shows that adding this familiarity buffer does indeed boost performance specially with rare events

**Summary Of The Review:**

This work tackles a complex problem and a very interesting research line. The method seems promising but the works is still incomplete with several points to be tackled for the community understand the solution presented in this work.

---

> ### Author Response · Authors · 2022-11-16
> **Response to Reviewer cQdQ**
>
> Thank you for the detailed review and thoughtful feedback. We provide our responses below.
>
> **Q1: the "method" section (Sec. 4), which is sometimes disjointed and could also improve with giving some intuition at some points.**
>
> We have now updated section 4 and have also added a paragraph at the beginning of the section to give an overview of the proposed approach. We hope the section is more clear now.
>
> **Q2: It only contrast their extension of IMPALA+MEM against IMPALA and IMPALA+MEM, it would have been a great addition to compare with a completely different kind of memory-based solution like HCAM [1] to contrast.**
>
> Thanks for pointing this out. We shall include this in the next version of our paper.
>
> **Q3: It would really benefit from ablation experiments. This would let the community know how influential is each element of your proposed solution, also it would clarify how sensible it is to the different hyperparameters.**
>
> We have already included the ablation results in the supplementary section of the paper. The authors request the reviewer to elaborate on what ''sensible" means.
>
> **Q4: End of page 3, it should give a reference to the contrastive loss since it hasn't been introduced yet**
>
> The equation (Equation 6) where this is introduced has been referred to in the updated version.
>
> **Q5: Page 4 line 3, "the augmented image" hasn't been explained yet**
>
> This has already been explained in the next immediate lines. Please let us know if we have misunderstood something.
>
> **Q6: It would be good to give some intuition there on why are you augmenting the image by adding noise and cutting a region**
>
> The explanation has now been added to the updated version of the paper.
>
> > We use these two augmentations because they are simple and don’t affect the task. For example, using something like random conv [2] will change the color of all objects in the observation. This might result in the agent confusing it to be an image (observation) from some other trial.
>
> **Q7: In section 4.1 you explain the momentum Eq. 4 and the contrastive loss (Eq.5 and 6). But not how they are related, which makes it confusing**
>
> They aren’t related, hence we haven’t shown it in the paper. But both are calculated from the same input (pixel embeddings), which is stated by the equations in that section.
>
> **Q8: Eq.7, you never explain what is v there**
>
> The 'v' in the paper was a typo and has now been changed to 'h', representing the hidden state of the LSTM.
>
> **Q9: Additionally, why using KNN and not the classic query-key match with hard attention? Hard attention is classically used in that context so would be good to motivate why authors did differently**
>
> We kept the architecture for fetching the hidden states the same as the baseline IMPALA+MEM [3]. Yes, it would be a good idea to see how hard attention works for this part of the architecture. We shall try this and add it in the next version of the paper as ablation.
>
> [1] Lampinen, Andrew, et al. "Towards mental time travel: a hierarchical memory for reinforcement learning agents." Advances in Neural Information Processing Systems 34 (2021): 28182-28195.
>
> [2] Laskin el al. "Reinforcement Learning with Augmented Data"
>
> [3] Fortunato et al., "Generalization of Reinforcement Learners with Working and Episodic Memory."

---

> > ### Comment · Reviewer_cQdQ · 2022-11-20
> > **Thanks and Response**
> >
> > I first want to thank authors for taking the time to responding to every question/concern of all reviewers and for the multiple updates in the work. I follow up with some questions:
> >
> > *Q1: Yes I think that the new intro and clarification comments in the text and pseudocode do help a lot, my only remaining point is about Q7 which I clarify below.
> >
> > *Q3: Silly mistake, I meant sensitive. And yes I was looking for something like what you have in Fig.6, I didn't see it first time. To help future readers not to miss that as I did, it might help  referring to the ablation experiments for the hyperparameters explicitly (right now you just mention ablations globally). And also, may be good to refer to the ablations at the end of the experiments section. At the beginning of the section the reader doesn't now yet it they would like to see additional experiments to those in the main body.
> >
> > *Q7: What I meant is that equations 5 & 6 specify the loss your method is optimizing, but just before you are explaining the momentum and average momentum loss. Thus it would help a lot how these different parts integrate together (maybe by refering to the algorithm and giving some written grounding on how these parts interact).
> >
> > The line of work is quite interesting I'll be keen to read the final version.

---

> > > ### Author Response · Authors · 2022-11-27
> > > **Response to Reviewer cQdQ**
> > >
> > > We thank the reviewer for reading the new version of the paper and are glad to know that the new intro and comments have improved the quality of the paper.
> > >
> > > We provide our response for further queries below:
> > > - Regarding the ablations, we have now added a few lines in the main paper to make the reader aware of the various ablations results added in the supplementary section.
> > > - We will make the equations and interaction between them more clearer in the body of the paper and include it in the final version of the paper.
> > >
> > > Please let us know if you have any more queries regarding anything. Happy to know that you find the work interesting! We are looking forward to you reading the final version.

---

### Official Review · Reviewer_uaJ9 · 2022-10-24

**Confidence:** 4
**Correctness:** 3
**Technical Novelty And Significance:** 2
**Empirical Novelty And Significance:** 2
**Recommendation:** 3

**Clarity, Quality, Novelty And Reproducibility:**

The writing is not clear. The method is incremental and lacks details to reproduce (see details in weakness)


**Strength And Weaknesses:**

### Strength:
Encouraging rare events in episodic memory is a reasonable idea. It fits well with the problem the paper aims to solve: Zipfian-like environments. Using contrastive learning to discover rare samples is well-motivated by long-tail learning literature. The experimental results show apparent performance gain in the 20% rarity case. Some baselines are wisely chosen to verify the contribution of the proposed idea.

### Weakness:
The method is relatively incremental. It merely combines components such as IMPALA, episodic memory and contrastive learning. The new module is the familiarity buffer, whose design is not surprising given the paper's main idea (storing the rarest events). This idea is not new either and can date back to this paper [1].

Another problem is the writing and presentation. The introduction provides little information on the operation of the proposed method, which creates a gap in understanding the claimed contributions. The notations are confusing and sometimes overlapping. The algorithm is hard to read; one may not implement this algorithm to reproduce the method.

Last but not least, the experiment is weak. Good results in a toy task and for only one setting (10 maps+10 objects) are not enough to validate the method. There are many additional hyperparameters. Since the performance changes as they vary (Table 2-5), it may require a lot of tuning to make this method work.

### Detailed comments and questions:

- The second paragraph in the introduction seems to be about other papers (Sutton & Barto, 2018b; Espeholt et al., 2018). Please rewrite it to clarify your contribution.
- Symbols like $tk$ can be written as $t_k$ for readability. T is used in both Eq. 2 and 3; are they different? What is N in Eq.4? What is v in Eq. 7?
- Sec. 4.2: IMPALA+MEM part should be described in detail in the background.
- The literature review is missing. Your references (e.g. Pritzel et al. (2017); Blundell et al. (2016)) are not up-to-date. It is better to have a separate section covering recent advances in episodic reinforcement learning.
- Algorithm: Please link the functions to the section you describe them.
- Fig. 1:  There should be k_t in the KNN part
- Fig. 4: There are some unexplainable cases. d) Contrastive learning can solve the rarest one (9-9) while your method fails. All methods fail in (5-1), which is far from the hardest. Why?
- Table 1: Why is your method better in the Uniform case? Is there any advantage?
- Hyperparameter: Does changing the memory/familiarity buffer size matter?
- To strengthen the experiment, please consider the following:

   * Test in other environments that also have rare events or sparse rewards. Many Atari and Minigrid environments exhibit these properties. Although they may not be precisely Zipfian, the need for long-tail discovery remains.
   * Test in different configurations of your toy environment by varying the environment's hyperparameters
   * Compare your contrastive learning with simple long-tail discovery methods, such as using VAE for novelty/outliner detection.
   * Consider some other episodic control methods as the baselines.

[1] Frank, Jordan & Mannor, Shie & Precup, Doina. (2008). Reinforcement learning in the presence of rare events.

**Summary Of The Paper:**

The paper proposes a mechanism to keep long-tail events in episodic memory to improve RL performance in Zipfian-like environments. Through contrastive learning, observations' rarity score is estimated and stored in a familiarity buffer. After certain training epochs, only the top rarest events from the buffer transit to the episodic memory, providing long-term context to the policy network following IMPALA architecture. The proposed method is tested in Zipf's Gridworld task and outperforms several variants of the IMPALA baseline, especially in handling long-tail maps.

**Summary Of The Review:**

Although the motivation of the paper is good, its method is incremental and the result is insufficient. The writing needs major improvement.

---

> ### Author Response · Authors · 2022-11-16
> **Response to Reviewer uaJ9**
>
> We thank the reviewer for providing valuable feedback to improve the work. We will address concerns and questions from the reviewer and answer them individually.
>
> **Q1: Symbols like tk can be written as t_k for readability.**
>
> We have updated the paper. tk has now been replaced by t_k, and tf has been replaced by t_f.
>
>
> **Q2:  T is used in both Eq. 2 and 3; are they different?**
>
> Yes, they are different. The T (Now Tr) in Equation 2 represents the trajectory of states, and the one in Equation 3 represents the number of epochs for which the contrastive loss is tracked.
>
> **Q3: What is N in Eq.4? What is v in Eq. 7?**
>
> N is the number of samples present in the familiarity buffer. A line has been added below to explain the same. Also, the 'v' in the paper was a typo and has now been changed to 'h', representing the hidden state of the LSTM.
>
> **Q4: Algorithm: Please link the functions to the section you describe them.**
>
> Comments have now been added in the algorithm to point to sections where the functions are described.
>
> **Q5: Fig. 1: There should be k_t in the KNN part**
>
> Can you please elaborate on this ? We aren't sure what k_t is being referred to here.
>
> **Q6: Fig. 4: There are some unexplainable cases. d) Contrastive learning can solve the rarest one (9-9) while your method fails. All methods fail in (5-1), which is far from the hardest. Why?**
>
> - (9-9) The theoretical probability of occurrence of that trial is 0.000041636. As you can see, the probability is very low for this trial, so during the experimentation (training), it might be possible that the agent has seen very few frames of that trial, even to explore the grid.
> - (5-1) Yes. the trial is far from the hardest in terms of probability of occurrence, but when we analyzed the trial, we noticed that the target object is placed very from the initial position (Image 1). Hence, the trial turns out to be harder than usual for the agent to do. The agent not only faces the problem of experiencing the trial rarely but also has to explore and find the target object that is very far from the initial position, making it even harder.
>
> Image 1: [9_9_anonymous.png](https://postimg.cc/CBkChMZX)
>
> **Q7: Table 1: Why is your method better in the Uniform case? Is there any advantage?**
>
> Our method can solve additional ‘rare’ trials that other methods (IMPALA, IMPALA+MEM, etc.) cannot solve. Hence, when evaluated on trials uniformly sampled, the accuracy is more.
>
> **Q8: Hyperparameter: Does changing the memory/familiarity buffer size matter?**
>
> Our intuition behind choosing 1024 as the buffer size was to keep the architecture as similar as possible to the MEM introduced in [1]. Hence, we have kept the size of both buffers equal (i.e., 1024). We did some minor ablations and didn’t see much difference in the results.
>
> **Q9: Test in other environments that also have rare events or sparse rewards. Many Atari and Minigrid environments exhibit these properties. Although they may not be precisely Zipfian, the need for long-tail discovery remains.**
>
> The main reason we used Zipf's gridworld is that it tests the agent on extreme conditions where the experiences can be very skewed. For example, the rarest 20% trials occur with a probability of less than 0.00006346. Also, the skewness in Atari Games is not that pronounced, hence it won’t be fair the judge the agents based on the metrics that they use (there won’t be much difference in the accuracy). We shall include some results on Atari Games in the future updated version of the paper.
>
> **Q10: Compare your contrastive learning with simple long-tail discovery methods, such as using VAE for novelty/outliner detection.**
>
> We tried an architecture where we trained a VAE on the observed states for novelty/outlier detection, but the results were unexpectedly poor, so we didn't want to include it in the paper.
>
> **Q11: Consider some other episodic control methods as the baselines.**
>
> Thanks for pointing this out. We shall report results on other episodic control methods in the next version of the paper.
>
> [1] Fortunato et al., "Generalization of Reinforcement Learners with Working and Episodic Memory."

---

> > ### Comment · Reviewer_uaJ9 · 2022-11-21
> > **Experiments still need improvement**
> >
> > Thank you for answering most of my questions. The revision looks better regarding presentation and clarity. It would be even better if you could include discussions such as those in Q6, Q8,  Q10 in the paper to make the paper more comprehensive.
> >
> > The experiments remain weak. It is critical to have more experiments and baselines to convince the readers about the benefit of the proposed method.
> >
> > Other notes:
> > - Q5. k_t  should be the key at timestep t that you use to search for the top-K nearest neighbour  k_i in the memory
> > - Q7. It is still unclear why the uniform setting can offer additional rare events. It would be beneficial to give some examples of the possible rare events in the uniform setting.

---

> > > ### Author Response · Authors · 2022-11-27
> > > **Response to Reviewer uaJ9**
> > >
> > > Yes, we plan to include those discussions in the final version of the paper. We thank the reviewer for taking a second look at the draft and are glad to know that the presentation and clarity are much better now. The additional experiments are running, and we plan to include those results in the final version of the paper.
> > >
> > > Response to other queries:
> > > - Thanks for the more precise query. We will include the k_t in the figure to clarify it even more.
> > > - We guess there is some misunderstanding in this part. We **train** our model on the environment where the trials are chosen according to the Zipfian distribution. During the uniform **evaluation**, we select the trial uniformly randomly. Hence each trial has an equal probability of appearing. Our model, unlike other models like IMPALA, IMPALA+MEM, etc., can detect rare states and prioritize during training, and hence it's able to solve additional trials on average that other methods aren't able to solve. For example, you can see in Figure 4 that trials like (2,4) and (3,4) are not solvable by most methods, but our method can solve it, and during uniform evaluation, our model tends to be scored higher.

---

### Official Review · Reviewer_h8Xx · 2022-10-24

**Confidence:** 4
**Correctness:** 2
**Technical Novelty And Significance:** 3
**Empirical Novelty And Significance:** 2
**Recommendation:** 3

**Clarity, Quality, Novelty And Reproducibility:**

Clarity:
- Medium. The main idea/algorithm is fairly clear when described in words, but the figures do not do a very good job of illustrating it. In particular, my suggestions for Figure 3 are to illustrate the merge/fetch operations better, as well as the "transfer tk rarest states" step.

Quality:
- Low. As noted above, the experimental evaluation is not sufficient. My suggestions would be to use the open-source Zipfian Gridworld task as described in the original paper, and add the other two tasks from [1] as well.

Novelty:
- Medium-High. The method appears novel and I think does hold promise, it just needs to be better demonstrated through the experimental evaluation.

Reproducibility:
- Low. The authors do not mention making their code public and do not include it.

**Strength And Weaknesses:**

Strengths:
- The setting is interesting and the general idea of considering heavily skewed task distributions is well-motivated
- The method does seem to show improvement, although it's only evaluated on a single task

Weaknesses:
- The main weakness of this paper is the experimental evaluation. First, only a single environment is used - this is insufficient to showcase the generality of the method. It could be that this particular approach is overfit to this task. I would suggest the authors test their methods on several more environments.
- For some reason, the authors do not use the original version of the Zipfian Gridworld task from [1] and make their own modified version of it (which is also easier, 10 objects/tasks instead of 20). They claim in their paper that this is because the code of [1] is not available, but it is: https://github.com/deepmind/zipfian_environments. It has been available long before the ICLR submission deadline. Since the environments from [1] are open source, I think they should be used instead to facilitate consistency across publications and to ensure comparability of results.

**Summary Of The Paper:**

This paper proposes a method for training RL agents in settings where the environment distribution is long-tailed, i.e. an agent must solve N tasks, but some of them occurs much more frequently than others. Examples of such a setting include the Zipfian RL environments introduced by [1]. This paper proposes a method that uses contrastive learning to get a familiarity score based on the contrastive loss (smoothed using momentum). This familiarity score is then combined with an episodic memory to provide additional learning for trajectories that belong to rare tasks.

The method is evaluated on a single environment, which is a modified version of the Zipfian gridworld introduced in [1]. It shows improvement over standard IMPALA, as well as different ablated versions of the proposed method, when the distribution of tasks is uniform rather than long-tailed, and the improvement is especially pronounced for rare tasks.








[1] Chan et al., "Zipfian environments for Reinforcement Learning".

**Summary Of The Review:**

Overall, I think this paper proposes a promising idea, but in its current form it is not ready for publication at ICLR. I would highly encourage the authors to extend the empirical evaluation as described in my comments. If the improvements reported here hold up to a stronger empirical evaluation, I think this would make a nice paper at a top conference.

---

> ### Author Response · Authors · 2022-11-16
> **Response to Reviewer h8Xx**
>
> We thank the reviewer for the valuable feedback and for suggesting additional insightful experiments. We provide our responses below.
>
> **Q1: only a single environment is used - this is insufficient to showcase the generality of the method.**
>
> There were 2 more tasks introduced in [1], but we didn't consider them for the following reasons:
> - The Zipf’s Labyrinth task tests the ability of the agents to learn in a meta-learning setup. Our problem is focused on learning in a single environment where the experiences are skewed.
> - The Zipf’s playroom tests the ability of the agent to learn in an environment where the agent is supplied with a string instruction, and it has to lift/put the corresponding object. This task mainly focuses on the ability to map language input to the corresponding objects in a skewed setting. Our architecture doesn’t take language as input but takes the instruction (target object to choose) along with the pixel observation (i.e., the target object is embedded in the RGB input).
>
> We could try this on some Atari Games, but we haven’t seen any such task where this long-tailed skewness is visible and well-highlighted. The main reason we used Zipf's gridworld is that it tests the agent on extreme conditions where the experiences can be very skewed. For example, the rarest 20% trials occur with a probability of less than 0.00006346. The skewness in Atari Games is not that pronounced, hence it won’t be fair the judge the agents based on the metrics that are used there (there won’t be much difference in the accuracy).
>
> We can design some new long-tail tasks and check the performance of our method on them, maybe in a 3D setting like Psychlab. We shall plan this for the next iteration of our paper.
>
> **Q2: For some reason, the authors do not use the original version of the Zipfian Gridworld task from [1] and make their own modified version of it (which is also easier, 10 objects/tasks instead of 20).**
>
> The code for the environments introduced in [1] wasn't available online when we started doing our experiments. It got released when we were midway through our ablation experiments. We didn't want to wait for the code release and hence had to resort to implementing our own version of this task.
>
> We ran all the experiments on a server that supports the spawning of a maximum of 50 actors (Shown in Supplementary Section A.3) in parallel. Due to this computational constraint, we worked on a smaller environment (10 maps and 10 objects). We can't say that this task is easier than the setting with 20 maps and 20 objects because they [1] have used 256 actors in parallel, whereas we are using just 50 actors to solve the task with 10 maps and 10 objects.
>
> Now that the original gridworld environment is available, it's not fair if we don't do experiments on that. Hence, we have now reported results for that as well (Supplementary Section A.1) in the updated version of the paper. We would like to thank the author for pointing this out.
>
> **Q3: my suggestions for Figure 3 are to illustrate the merge/fetch operations better, as well as the "transfer tk rarest states" step.**
>
> We have updated Figure 3 in the updated version. A ⨁ has been added to illustrate the fetch and merge operation, which is further explained in the caption.
>
> [1] Chan et al., "Zipfian environments for Reinforcement Learning."

---

> > ### Comment · Reviewer_h8Xx · 2022-12-09
> > **Thanks for the updates**
> >
> > Thank you for the additions and for running the experiments on the original Zipf's gridworld and for updating the figure. I see the proposed method still offers improvements, which is good. However, I still think that testing a method on a single gridworld environment is not enough for publication, even though the initial results are encouraging.
> >
> > Some ideas for the next version of the paper which would strengthen the experiments could be:
> > - train an agent on the Atari suite with all the games combined (i.e. at each episode, a random game is sampled), but weight the different games according to a Zipfian distribution. That is, some games will show up frequently while others will show up very rarely. We could then see how the different methods do on the rarer games.
> > - Do the same for some procedurally generated games, such as ProcGen  or MiniHack. Fix a set of seeds, and at each episode a seed is sampled according to a Zipfian distribution, and that seed is used to generate the environment. This way, some seeds will occur much more frequently than others, and we can see how the methods do on the rarer seeds.
> >
> > I would encourage the authors to think about different ways they can test their method on more environments and add more experiments to the paper, in which can I think it will be a strong submission.

---

### Decision · Program_Chairs · 2023-01-20

**Decision:**

Reject

**Justification For Why Not Higher Score:**

Reviewers all agreed that the experiments were too limited by being focused on only one environment. Lack of clarity was also a common issue brought up, indicating that this work in its current form is not ready for publication at ICLR.

**Justification For Why Not Lower Score:**

N/A

**Metareview: Summary, Strengths And Weaknesses:**

In this paper, the authors tackle the challenge of learning in RL environments with long-tailed distributions (tasks are not sampled iid but according to a Zipfian distribution). They propose a method which uses contrastive learning to determine how rare a state is in order to determine whether to store it in episodic memory. They tested on a variant of Chan et al’s Zipfian gridworld tasks and found improvement over standard IMPALA.

Reviewers found it a well-motivated and reasoned idea, and the results on the presented environment are generally impressive. However many expressed major concerns that only a single environment was used, which limits the generalizability of their findings. Other issues brought up were that only a single type of memory agent was compared, and that the methods and overall presentation are a bit unclear.

During the rebuttal period, the authors did update their paper to test on the original Zipfian gridworld environments, which had since been open-sourced, but this unfortunately didn’t satisfactorily address reviewers’ concerns about the limited nature of their tasks. One reviewer suggested converting existing environments such as the Atari suite, ProcGen, or Minihack into Zipfian ones by sampling from them under a long-tailed distribution, which I think is a great idea that would significantly enhance the quality of their experiments.

Reviewers were relatively engaged with discussions, which I think were overall fruitful and helped to improve the paper’s clarity. While I cannot recommend acceptance at this time, I hope their comments will be helpful for future iterations of this work and I look forward to future iterations of it.